

**Uptake Behavior of Polycyclic Aromatic Compounds during Field**
**Calibrations of the XAD-Based Passive Air Sampler Across Seasons and**
**Locations**
Yuening Li,[1] Faqiang Zhan,[1,*] Yushan Su,[1,#] Ying Duan Lei,[1] Chubashini Shunthirasingham,[2]
Zilin Zhou,[3] Jonathan P. D. Abbatt,[3] Hayley Hung,[2] Frank Wania[1,*]
[1]*Department of Physical and Environmental Sciences, University of Toronto Scarborough,*
*1265 Military Trail, Toronto, Ontario, Canada M1C 1A4*
[2]*Air Quality Processes Research Section, Environment and Climate Change Canada, 4905*
*Dufferin Street, Toronto, Ontario, Canada M3H 5T4*
[3]*Department of Chemistry, University of Toronto, 80 St George Street, Toronto, Ontario,*
*Canada M5S 3H6*
[#]current: *Ontario Ministry of the Environment, Conservation and Parks, 125 Resources Road,*
*Etobicoke, Ontario, Canada M9P 3V6*
[*]Corresponding authors: frank.wania@utoronto.ca, faqiang.zhan@utoronto.ca



## ABSTRACT

Polycyclic aromatic compounds (PACs) continue to demand attention due to their widespread presence and well-established health implications. Given that incomplete combustion is a major contributor to PACs and inhalation constitutes a crucial human exposure pathway, a comprehensive understanding of the concentrations, spatial distributions, and fates of a broad range of PACs in the atmosphere is important. Passive air samplers (PASs) are a commonly utilized technique for PAC sampling and monitoring. In this study, we present the results from two one-year calibration experiments, one starting in summer and the other in winter, using a passive air sampler equipped with XAD resin as the sorbent (XAD-PAS). Throughout both experiments, PACs were consistently sorbed during the initial six-month period. However, the sorbed amounts for many PACs exhibited a decrease after half a year of deployment. Three hypotheses to explain this phenomenon were explored, including the uptake of atmospheric particles, evaporation from the sorbent, and reactions with photooxidants. All had to be rejected based on the obtained data, additional laboratory experiments and model results. Model simulations were further used to (i) confirm that a loss process must be invoked to explain the observed uptake behaviour and (ii) estimate the kinetics of that loss process for different PACs. Sampling rates ($SR$s) for 28 PACs derived from the linearized uptake curves during the first six months of deployment were comparable to those of other semi-volatile organic compounds obtained during the same calibration experiment, and they also demonstrate a consistent negative correlation with volatility.



## 1. INTRODUCTION


Approximately 4 million square kilometers of savanna, forest, grassland, and agricultural
ecosystems are burnt through both natural and controlled fires annually (Nolan et al., 2022).
These fire events have attracted global attention to the release of pollutants, including harmful
particles, organic vapors, and greenhouse gases, into the air as well as the related potential
health risks. In particular, the production and dispersion of polycyclic aromatic compounds
(PACs) in the environment have emerged as significant concerns. PACs comprise organic
molecules composed of fused aromatic rings, including unsubstituted polycyclic aromatic
hydrocarbons (PAHs), alkylated PAHs (alk-PAHs), and heterocyclic aromatic compounds
containing N, O, or S atoms in their structure (Moradi et al., 2022). PACs originate from a
multitude of sources, encompassing both natural and human-related activities, many related to
the incomplete combustion of organic matter. Natural sources include wild fires (Nolan et al.,
2022; Environment Canada and Health Canada, 1994; Nikolaou et al., 1984; Wnorowski et al.,
2021), volcanic eruptions (Nikolaou et al., 1984; Programme United Nations Environment,
2020), and biogenic processes (Wakeham et al., 1980). Anthropogenic activities contribute
substantially to the emissions of PACs into the environment (Environment Canada and Health
Canada, 1994), with vehicle emissions (Berthiaume et al., 2021; Muir and Galarneau, 2021),
cooking and heating (Shen et al., 2012; Environment Canada and Health Canada, 1994), wood
burning (Lima et al., 2005; Xu et al., 2006), industrial processes (Lima et al., 2005; Xu et al.,
2006), and tobacco smoke (Holme et al., 2022) releasing numerous PACs into the atmosphere.
PAHs have been widely studied due to their ubiquitous presence in the environment and their
potential for eliciting adverse health effects, such as allergic potential, carcinogenicity,
teratogenicity, and genotoxicity (Rice and Baker, 2007; Boffetta et al., 1997; White, 2002; Kim
et al., 2013; Organization, 1998; ATSDR (Agency for Toxic Substances Disease Registry),
1995), PAHs are regulated and routinely monitored by numerous agencies and governments
across the globe.
With past research efforts mainly focused on PAHs, alk-PAHs have only recently started to
garner attention from researchers. The addition of alkyl groups alters the physicochemical
properties of PAHs, affecting their solubility, volatility, and toxicological properties. Certain
alk-PAHs may be more toxic than their non-alkylated counterparts (Golzadeh et al., 2021;
Hawthorne et al., 2006; Andersson and Achten, 2015; Grung et al., 2011; Kaisarevic et al.,
2009; Pcchillips et al., 1979; Sarma et al., 2017), raising concerns about their potential
ecological and human health impacts. Whereas alk-PAHs have been identified at elevated



concentrations, and in some cases, are even dominant among PACs in air (Wnorowski et al.,
2022; Moradi et al., 2022; Jariyasopit et al., 2019) and food (Golzadeh et al., 2021), they have
not been widely studied, monitored, and regulated (Moradi et al., 2022). Studies in Canada are
particularly limited in number, with some data available for the Greater Toronto Area
(Jariyasopit et al., 2019; Moradi et al., 2022) and Athabasca oil sands region (Harner et al.,
2013; Jariyasopit et al., 2018; Wnorowski et al., 2021; Mahoney et al., 2023; Cheng et al., 2018;
Rauert et al., 2020; Harner et al., 2018; Moradi et al., n.d.; Jariyasopit et al., 2021; Ahad et al.,
75  2021).

Because inhalation is a critical human exposure pathway (Carl-Elis et al., 2002; Liu et al.,
2007), achieving a comprehensive understanding of PAC concentrations, spatial distribution,
and fate in the atmosphere is important. As atmospheric PAC concentrations are related to
proximity to emission sources, urban areas, industrial zones, regions with high traffic density,
and places close to wildfire typically exhibit elevated PAC concentrations. Atmospheric
transport can disperse PACs widely, both while bound to particles and in the gas phase (Muir
and Galarneau, 2021; Zhou et al., 2019; Wnorowski et al., 2022; Masclet et al., 2000). Clearly,
there is a need for reliable air sampling techniques for a wide range of PACs.
Atmospheric PACs can be sampled using active air samplers (AASs) and passive air samplers
(PASs). In AASs, pumps are used to pull air through a sampling medium (e.g., a sorbent or/and
a filter) to capture atmospheric PACs in the gas or/and particle phase. While accurate sampling
volumes are usually easily obtained, the need for a stable electrical power supply and high
maintenance requirements and operational expenses limit the geographical scope of AASs,
especially in remote areas. Without using pumps, PASs sample and retain chemicals by relying
on chemical vapors' diffusing and sorbing to a sorbent. Low cost and maintenance
requirements expand their potential spatial applications, e.g., in areas close to wildfire regions.
However, obtaining accurate sampling volumes can be challenging. Confidently using a PAS
requires quantitative knowledge of the uptake kinetics and of the limits of linear uptake for the
targeted compounds, which is typically obtained by calibration studies using co-located AAS
and PASs.
Sorbents used for sampling PACs in PASs include polyurethane form (PUF) (Cheng et al.,
2013; Domínguez-Morueco et al., 2017; Pozo et al., 2015), polyethylene (PE) (Bartkow et al.,
2004; Meierdierks et al., 2021), polydimethyl siloxane (PDMS) (Barthel et al., 2012; Bohlin-
Nizzetto et al., 2020), and styrene−divinylbenzene co-polymeric resin (commercial name XAD)
(Barthel et al., 2012; Lévy et al., 2018). Calibration studies have been conducted for PACs in



PAS based on PE (Meierdierks et al., 2021), PUF (Harner et al., 2013; Bohlin-Nizzetto et al.,
2020; Tromp et al., 2019; Holt et al., 2017; Bohlin et al., 2014a; Melymuk et al., 2011), PDMS
(Tromp et al., 2019), and XAD(Ellickson et al., 2017; Armitage et al., 2013). However, due to
a relative low uptake capacity, some sorbents, such as PE and PDMS, are less widely used in
PAS. Even though PUF-PASs are widely used, they have some limitations: (1) the uptake
capacity of PUF is too small to sample more volatile PACs within the linear uptake regime
during longer deployment periods, which may lead to relatively high uncertainties and
difficulties in data interpretation (Li and Wania, 2021; Li et al., 2022). (2) Gas-phase
concentrations of PACs, especially of less volatile PACs, may be hard to obtain as the PUF-
PAS samples both gaseous and particle-bound PACs (Melymuk et al., 2011), with sampling
rates for the latter suffering from high variability and uncertainty (Holt et al., 2017).
The significantly greater uptake capacity (Wania and Shunthirasingham, 2020; Hayward et al.,
2011) of the XAD-PAS (Wania et al., 2003) results in a demonstrably longer linear uptake
period (Wania and Shunthirasingham, 2020; Li et al., 2023a, b) for many semi-volatile organic
compounds (SVOCs) compared to other PASs. Moreover, the XAD-PAS's shelter was
intentionally designed to mitigate the impact of wind and particle uptake. Although the XAD-
PAS has been calibrated for PAC twice (Ellickson et al., 2017; Armitage et al., 2013), these
studies only targeted a limited number of PAHs and no alk-PAHs or heterocyclic aromatic
compounds. While Armitage et al. (2013) deployed AASs and PASs side-by-side, Ellickson et
al. (2017) positioned certain PASs at a considerable distance from the AASs, which could
potentially introduce heightened levels of uncertainty into the findings. The current study
sought to conduct a "gold standard" calibration for XAD-PAS with the following objectives:
(1) evaluate the uptake behaviors of a large number of PACs in XAD-PAS, including PAHs,
alk-PAHs, and the heterocyclic aromatic compound dibenzothiophene; (2) determine the linear
uptake regimes and obtain experimental sampling rates for these PACs; (3) assess the impact
of meteorological conditions and chemical properties on the sampling of these PACs.
**2. METHODS AND MATERIALS**
**Field Sampling.** The main calibration experiment was conducted in 2020/2021 on the campus
of the University of Toronto Scarborough in the eastern suburbs of Toronto (43.7837, -
79.1903), with results for other SVOCs reported previously (Li et al., 2023b, a). For
comparison, we also present the results of calibration experiments conducted in 2001/2002 in
a forest (44.3184, -79.9341) and a nearby clearing (44.3270, -79.9169) site in Borden, Ontario,
with air concentrations having been previously reported (Su et al., 2007a, b). The Borden sites



are approximately 85 km to the Northwest of the Toronto site. In each study, XAD-PASs were
deployed simultaneously ca. 1.5 m above the ground, and AASs, collecting particle and gas
phase separately, were placed side by side with the PASs. Both calibration experiments lasted
~1 year, with the Borden experiments starting in November 2001 and the Toronto experiment
in June 2020. During the experiments, XAD-PASs were retrieved every four weeks in Toronto
and after 36, 60, 120, 181, 246, 323, and 365 days in Borden. Periodical 24-hour-long active
air samples in Borden and consecutive week-long active air samples in Toronto were obtained
using high-volume active air samplers (HV-AASs) and mid-volume active air samplers (MV-
AAS), respectively. Detailed sampling information has been described in previous publications
(Su et al., 2007a, b; Li et al., 2023b, a).
**Sample Treatment.** Glass fiber filters (GFFs) and PUFs from the Borden experiment were
Soxhlet extracted with dichloromethane and petroleum ether, respectively. Extracts were
cleaned, fractionated, and concentrated as described previously (Su et al., 2006, 2007a). The
XAD from the PASs deployed at Borden was loaded to and extracted in an elution column with
250 mL of methanol, followed by 350 mL of dichloromethane. After removing methanol with
250 mL of 3% sodium chloride, extracts were concentrated and fractionated using the method
described in Wania et al. (2003) The fractionated extracts were solvent exchanged into iso-
octane, and finally concentrated to 0.5 mL using nitrogen. More detail on sample treatment is
available in earlier publications (Su et al., 2006, 2007a; Wania et al., 2003).
For the samples from the Toronto experiment, one XAD-PAS cylinder from each retrieval date,
all blank samples, the PUF-XAD-PUF sandwiches, and GFFs were subjected to extraction
using an accelerated solvent extractor (Dionex 350). Prior to extraction, labeled standards
(Table S2) were spiked onto the samples as surrogates. The extracts were then concentrated
using rotary evaporation, water residues were removed using sodium sulfate columns, solvent-
exchanged into iso-octane, and further concentrated to 0.1 mL for PAS and 0.5 mL for AAS
using nitrogen-blowdown. Prior to instrumental analysis, 2.96 ng and 14.8 ng injection
standards (Table S2) were added to PAS and AAS extracts, respectively. Detailed procedures
were reported previously (Li et al., 2023b, a).
**Instrumental Analysis.** Extracts from the Borden experiment were analysed for 16 PAHs with
an Agilent 6890 gas chromatograph (GC) equipped with an Agilent 5973 mass spectrometric
(MS) detector using electron ionization in selected ion monitoring mode. In extracts from the
Toronto experiment, a total of 22 PAHs, 22 alk-PAHs and dibenzothiophene were analyzed.
This analysis were carried out using an Agilent 7890A GC coupled with a 7000A triple



quadrupole MS for PAS and PUF/XAD sanwich extracts and and an Agilent 8890 GC coupled
to with a 7010B MS/MS for the GFF extracts, using electron ionization in multiple reaction
monitoring mode. Details on instruments, columns, temperatures, and other parameters are
given in the Supporting Information (SI).
**Ozone Exposure Experiment**. A laboratory experiment was conducted to examine the
potential of PAHs to undergo reactions with ozone while being sorbed to XAD. Approximately
150 g of pre-cleaned XAD-2 resin was added to ~300 mL of an acetone solution containing
~3500-5000 ng each of acenaphthene, anthracene, phenanthrene, pyrene, and fluoranthene. To
make sure the XAD-resin was well mixed, it was stirred continuously using a clean spatula
until all acetone had evaporated (confirmed by repeatedly weighing of the PAC-loaded XAD-
resin). ~5.0 g of the spiked XAD resin was added to 10 mesh cylinders (2.0 cm diameter), as
used in the XAD-PAS, reducing the amount of resin required by placing a smaller empty mesh
cylinder (1.0 cm diameter) in their centre (Zhang et al., 2011). The exact weight of the XAD
in each mesh cylinder was recorded and used for data normalization.
Five randomly selected spiked mesh cylinders were extracted and analyzed using the same
method as for the samples from the Toronto experiment mentioned above in order to investigate
the variability of the PAHs in these cylinders. Three mesh cylinders were exposed to 1000 ppb
ozone at 0% relative humidity (RH) in a flowtube (Zhou and Abbatt, 2021) (Figure S1), two
for 24 hours and one for 72 hours. As a control, two mesh cylinders were exposed to air for 24
hours at 0% RH in the same flow tube. Upon retrieval, the five mesh cylinders from the flow
tube experiments were extracted and analyzed as before.
**Calculation of Sampling Rates (*SR*).** During the linear uptake phase, the *SR* of a compound
in the XAD-PAS is linearly related to the effective sampling volume $V_{eff}$ (m$^3$):
$$V_{eff} = \frac{M_{\text{PAS}}}{C_{\text{air}}} = SR \cdot t \qquad (1)$$
where $M_{\text{PAS}}$ (ng) is the field blank corrected amount of a PAC sorbed to the XAD-2 resin, and
$C_{\text{air}}$ is the atmospheric gas phase concentration of the PAC (ng m$^{-3}$) averaged over the
deployment period *t* of a sampler (day). $C_{\text{air}}$ is derived by averaging the concentrations recorded
by the AAS during a PAS's deployment. The *SR* for each compound during the period of linear
uptake is then obtained as the slope of the linear regression between $V_{eff}$ and *t*.
**Chemical Properties Compilation.** The logarithm of the equilibrium concentration ratios
between XAD-resin and air ($K_{\text{XAD/air}}$, L air g$^{-1}$ XAD) at 20 °C (Table S4) and the internal





energies of phase transfer between XAD and the gas phase $\Delta U_{XAD/air}$ (J mol$^{-1}$, Table S4) were
calculated using solute descriptors for PACs from the UFZ-LSER database (UFZ-LSER
database v 3.2.1 [Internet], 2022) and the poly parameter free energy relationships (ppLFERs)
by Hayward et al. (2011) The log $K_{XAD/air}$ values at the mean temperatures over the entire
deployment period of the Borden and Toronto calibration experiments (7.9 and 12.2℃,
respectively) (Table 2) were obtained from log $K_{XAD/air}$ at 20 ℃ and $\Delta U_{XAD/air}$ using the van't
Hoff equation (Atkinson and Curthoys, 1978; Goss, 1996).
**Model Simulations.** A mechanistic model develped by Zhang and Wania (2012) was used to
simulate the uptake of PACs with different $K_{XAD/air}$ during the two calibration experiments
beginning in different seasons (June and November). This model describes the diffusion of
chemicals from the atmospheric gas phase to the sorbent of a PAS and the kinetics of reversible
sorption to that sorbent. It has been modified to consider degradation loss of chemical sorbed
to the XAD-resin and to incorporate time-variant temperature and atmospheric ozone
concentrations. The Levenberg-Marquardt algorithm was used to find the combinations of
sorption rate $k_{sorb}$ and degradative loss rate $k_d$ that provide the best fit between predicted and
measured uptake curves. The actual temperature measured during the two calibration
experiments (Figures S2 and S3) were used in the simulations. Measured ozone concentrations
reported for the vicinity of the sampling sites (Figures S4 and S5) were used as input for
photooxidant degradation simulations only. The thickness of the stagnant air boundary layer
was assumed to be 0.01 cm and the simulated deployment length was 336 days.
**3. RESULTS AND DISCUSSION**
**Amounts Accumulated in PASs.** Fifteen PAHs, twelve alk-PAHs and dibenzothiophene were
reliably detected in the PAS extracts from the Toronto experiment (Table S6). Seven of these
PAHs were also detected in the extracts from the Borden experiment (Table S7). The amounts
of fluorene, naphthalene, and phenanthrene accumulated in the PASs are plotted versus
deployment length in Figure 1. Plots for the remaining 25 PACs are provided in Figures S6 and
S7 in the Supporting Information. These plots reveal that for only five of the 28 PACs in the
Toronto experiment, namely, acenaphthylene, fluorene, 1-methylfluorene, chrysene, and
benzo[b]fluoranthene, does the amount sorbed to the XAD increase relatively continuously
with increasing deployment for the entire length of the experiment. The amounts of the
remaining PACs either were largely constant (e.g., naphthalene in Figure 1) or decreased (e.g.,
phenanthrene in Figure 1) after approximately six-months of deployment. The observed uptake
behaviour was consistent between the three study sites, i.e., was observed in Toronto and the



forest and clearing sites in Borden (Figure 1). This behaviour is in contrast to continuous uptake
observed for a large number of SVOCs targeted in the same calibration study in Toronto (Li et
al., 2023b, a).
Theoretically, the amount of a chemical taken up in a PAS should always increase with
deployment length (Wania and Shunthirasingham, 2020). A decrease with increasing
deployment length might occur if the chemical is degraded while being sorbed or if the rate of
evaporation from the sorbent exceeds the rate of uptake. The latter occurs only if a chemical's
air concentration decreases or temperature increases after the chemical has reached a state of
equilibrium between atmospheric gas phase and sorbent. Equilibrium is more easily established
for volatile chemicals, sorbents with low uptake capacity and long deployments. We explored
three hypotheses to explain these unexpected decreasing trends in the amounts accumulated in
PASs.
**Can Uptake of Particle-bound Compounds Explain Decreasing Trends in the Amounts**
**Accumulated in PASs?** The first hypothesis posits that the observed uptake behaviour was
caused by the potentially inconsistent and unpredictable uptake and accumulation of
atmospheric particles in the XAD-PAS. This hypothesis is based on previous studies showing
that some PASs do not efficiently block wind from carrying particle-bound substances to the
sorbent, as is, e.g., observed in the case of the PUF-PAS with double bowl shelters (Wania and
Shunthirasingham, 2020; Chaemfa et al., 2009; Bohlin et al., 2014b; Harner et al., 2013). Even
though there is currently no evidence of the uptake of particle-bound SVOCs in the XAD-PAS,
if such uptake were to occur, the accumulated amount of a chemical in the XAD-PAS would
be the sum of the amount taken up from the gas phase and the amount derived from sampled
particle phase. Consequently, the uptake amount could be influenced by the size, type, and
concentrations of particles in air and the wind exposure of a PAS during deployment.



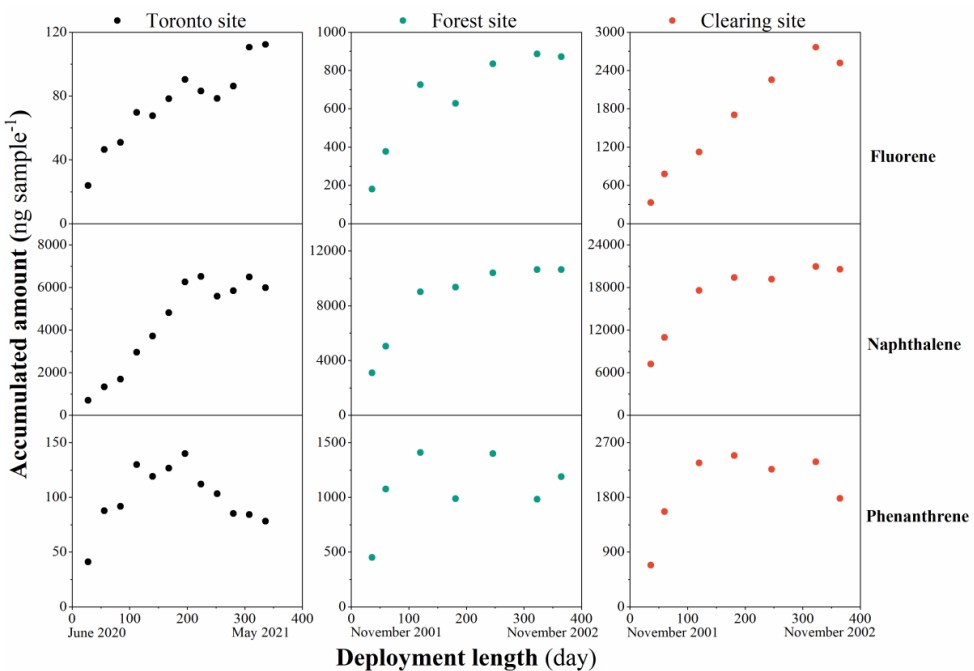

**Figure 1:** The amounts of three representative chemicals accumulating in the XAD-PASs throughout the one-year deployment period. The black, green, and red markers indicate the blank-corrected measured values at the site in Toronto, and the forest and clearing sites in Borden, respectively. The deployment month and the month in which the last PAS was retrieved are also indicated at the bottom part of this figure.

Seventeen PAHs and four alk-PAHs were reliably detected in the particles collected on the GFFs from the AASs in the Toronto experiment. Almost all two- and three-rings PACs were only detected in the gas phase, some four-rings and five-rings PACs could be detected in both gas and particle phase, and nine less volatile PACs were only found in the particle phase (Figure 2). This is consistent with previous studies (Lewis and Coutant, 2020; Ravindra et al., 2008; Terzi and Samara, 2004) indicating less volatile PACs with four or more rings mainly being associated with atmospheric particles, whereas volatile PACs are mostly in the gas phase.

Only PACs with at least four or more fused rings in their structure could be detected with a high percentage in the particle phase, whereas the chemicals taken up by the XAD-PASs were primarily more volatile PACs with three or fewer rings. The absence of these relatively volatile PACs in the particle phase (i.e., levels below the LOD) demonstrates that the variation of their accumulated amount in XAD-PASs in the second half of the year-long deployment cannot be associated with the uptake of atmospheric particles. Because concentrations of particle-bound




PACs in Ontario are much higher in winter than in summer (Su et al., 2007a), their hypothetical
uptake would be expected to occur during the latter half of the Toronto experiment and the first
half of the Borden study. However, the same deviations from continuous uptake are apparent
in the second half of both experiments (Figure 1).

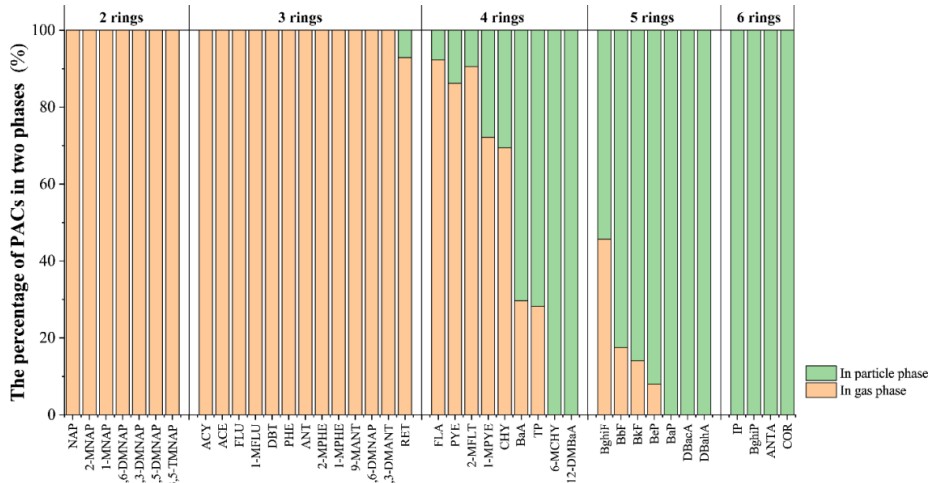


**Figure 2:** The average percentages of PACs detected in the particle (green) and gas phase (orange) of
the 48 week-long active air samples taken during the Toronto experiment. PACs were
grouped based on the number of rings in their structure.
In the case of the least volatile PACs with five and six rings that were only detected in the
particle phase, their absence in PAS extracts provides further evidence that the XAD-PAS does
not take up atmospheric particles. Daly et al. (2007) previously reported that involatile
chemicals tend to have levels below the limit of detection in XAD-PAS extracts, which is also
consistent with our previous studies indicating that no SVOCs with log $K_{\text{XAD/air}}$ higher than 7.0
at 12.2 °C were detected in XAD-PASs (Li et al., 2023b, a). All evidence therefore refutes the
first hypothesis that particle uptake may be the cause of the variation in the uptake amounts.
**Can a Limited Sorbent Uptake Capacity Explain Decreasing Trends in the Amounts**
**Accumulated in PASs?** The second hypothesis is that the uptake capacity for XAD-PAS is
too small to prevent some PACs from reaching equilibrium between sorbent and atmospheric
gas phase after six months of deployment. A subsequent decrease in sorbed amount could then
be attributed to a decrease in air concentration or an increase in temperature, both of which
would allow evaporative loss to exceed uptake. Several pieces of evidence do not support this
interpretation:



During the same calibration experiment in Toronto, we observed highly linear uptake over the
entire 48 weeks for SVOCs as volatile as hexachlorobutadiene and monochlorinated biphenyl,
with estimated log ($K_{XAD/air}$ / (L air g$^{-1}$ XAD)) values of 2.95 and 3.56 at 20 °C, respectively.
Naphthalene, the most volatile of the PACs targeted, has an estimated log ($K_{XAD/air}$ / (L air g$^{-1}$
XAD)) of 2.69. It is implausible that some chemicals remain in the linear uptake phase, while
chemicals with a very similar or higher affinity for sorbing to XAD from the gas phase would
reach equilibrium in the same experiment. Furthermore, there is no indication that the extent
of loss of PAC from the PAS during the second half of deployment is related to compound
volatility. For example, whereas phenanthrene is less volatile than either naphthalene or
fluorene, i.e., has a higher $K_{XAD/air}$ (Table 2), it was lost from the XAD-PAS to a greater extent
than either of these PACs (Figure 1).
The potential for evaporative net loss of a PAC from the XAD-PAS is largest in summer, when
PAC air concentrations tend to be lower and temperatures are highest. This implies that this
potential for evaporative loss during the second half of a one-year experiment would be higher
in the case of the Borden calibrations (starting in November) than in the Toronto experiment
(starting in June). Figure 3 displays simulated uptake curves for a compound that is volatile
enough to reach equilibrium between XAD and gas phase. Applying the seasonal variability of
naphthalene air concentrations and the temperatures measured in Borden and Toronto (Figure
S7), the model predicts widely divergent uptake curves for deployments starting in November
and June. The loss of sorbed naphthalene occurs earlier during an experiment starting in winter
(after ca. 120 days of deployment), and the extent of loss is more pronounced. In contrast, the
naphthalene uptake curves observed in the field calibration experiments are remarkably similar
(Figure 1). In summary, all evidence indicates that PACs did not reach equilibrium and a
limited XAD uptake capacity cannot explain the absence of continuous uptake.
**Can Degradative Loss Explain Decreasing Trends in the Amounts Accumulated in PASs?**
The third hypothesis posits that reactions of sorbed PACs with photooxidants present in the
atmosphere could account for the lack of continuous uptake (Jariyasopit et al., 2015; Melymuk
et al., 2017). It is well established that ozone can react with PAHs sorbed to solid phases
(Borrowman et al., 2016; Zhou et al., 2019) and atmospheric particles (Van Vaeck and Van
Cauwenberghe, 1984; Kasumba and Holmén, 2018). While nitrogen dioxide may also react
with some PAHs, either no reactions were found (Grosjean et al., 1983; Pitts et al., 1980), or
only negligible percentages of PAHs were observed to react with nitrogen dioxide (Tokiwa et
al., 1981). Even though gaseous and sorbed PAHs can react with OH radicals (Brubaker and





Hites, 1998; Atkinson and Arey, 2007; Esteve et al., 2004, 2006; Bedjanian et al., 2010), such
reactions are less likely to occur within the dark environment of a XAD-PAS housing.
Consequently, we focused solely on the reaction between ozone and PACs.

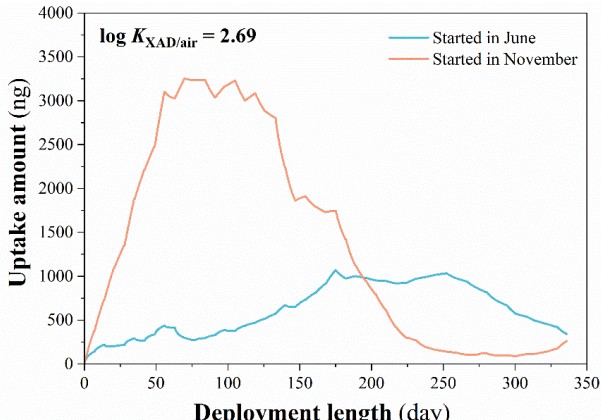


**Figure 3:** Predicted uptake curves for a chemical with a log ($K_{XAD/air}$ / (L air g$^{-1}$ XAD) value of 2.69 at

20 °C during calibration experiments starting in June and November, while assuming air
concentration and temperature variability typical for Southern Ontario. The thickness of the
stagnant air boundary layer was set to 0.01cm, the sorption rate was set as $3\times10^5$ day$^{-1}$ and
no degradation was assumed.
Specifically, we exposed PAC sorbed to XAD to high concentrations of ozone in a flow tube
to probe the possibility of degradative loss. If the reaction rate of the PACs with ozone is
assumed to be proportional to the ozone concentration, an exposure to ~1000 ppb of ozone for
24 and 72 hour is equivalent to two to four months of continuous exposure to atmospherically
relevant ozone levels of ~25 ppb. During the sampling period in Toronto, the ozone
concentration measured at an air monitoring station (43.7479, -79.2741) located ca. 8 km from
our sampling site ranged from 10 to 50 ppb with an annual average of 26 ppb (Figure S4). The
ozone concentration in the vicinity of the Borden site had a similar range (10 to 60 ppb) and
mean (27 ppb) (Figure S5).
We succeeded in obtaining a consistent loading of PAHs in the XAD-filled mesh cylinders:
The amounts of five PAHs on the XAD, when normalized to the net weight of resin in a mesh
cylinder, had a relative standard deviation (RSD) between 4.6 and 6.6% (Table 1). Neither
extended exposure to ozone in the flow tube nor exposure to air in the control experiment
resulted in a significant change ($p > 0.05$) in these amounts of PAHs in the XAD-filled mesh
cylinders (Table 1). In other words, the experiments suggest that ozone did not react with the



PACs sorbed to XAD resin. Interestingly, we did observe the continuous loss of ozone during
the flow tube experiments (Figure S8). This loss was much larger than when no XAD was
present in the flow tube, i.e., can be attributed to the resin. One potential explanation is that
ozone reacts with the benzene rings in the XAD resin and because of the orders of magnitude
higher abundance of these aromatic structures compared with the sorbed PAHs, this reaction
may protect the sorbed PAHs from being attacked.
**Table 1** The amounts of spiked PAHs on XAD resin before and after ozone exposure

| | The amount of PAHs on XAD resin (ng) | | | | |
|---|---|---|---|---|---|
| | Acenaphthene | Anthracene | Phenanthrene | Pyrene | Fluoranthene |
| **Loading Test** | 60.2 ± 3.1 | 84.1 ± 5.6 | 66.9 ± 3.1 | 120.3 ± 6.6 | 63.2 ± 3.4 |
| **Control Group** | | | | | |
| 0 ppb $O_3$ (1 day) | 64.5 ± 1.3 | 87.1 ± 8.1 | 68.1 ± 4.8 | 126.4 ± 2.6 | 65.5 ± 3.0 |
| **Experimental Group** | | | | | |
| 1000 ppb $O_3$ (1 day) | 61.7 ± 1.2 | 84.8 ± 2.5 | 66.4 ± 2.4 | 123.8 ± 3.5 | 64.8 ± 1.9 |
| 1000 ppb $O_3$ (3 days) | 62.5 | 91.3 | 68.7 | 126.9 | 67.0 |

Further evidence that reactions with photooxidants are not responsible for the loss of PACs
from the XAD-resin is provided again by the expected seasonal variability in the importance
of that process. In both Toronto and Borden, the ozone concentrations are seasonally variable
with higher levels in late summer and early fall and lower levels in winter (Figures S4 and S5).
It thus should affect the second half of a one-year deployment much more strongly in the
Borden experiment than the Toronto experiment. No such difference is apparent. To simulate
the influence of ozone on the uptake amount of a PAC, a model simulation was conducted for
three representative PACs in the Toronto experiment using the model by Zhang and Wania
(2012) modified to allow the sorbed PAC to react with atmospheric ozone. The results from
simulations show that the $R^2$ values of the best fitting uptake curves for these three PACs
decrease when the actual time-variant ozone concentration is used as an input parameter
(Figure S9). Finally, we note that PAHs that are known to have a higher reactivity with ozone
than others, such as fluorene (Kasumba and Holmén, 2018), anthracene (Kasumba and Holmén,
2018), and benz(a)anthracene (Van Vaeck and Van Cauwenberghe, 1984), do not appear to
show a higher rate of loss from the XAD-PAS when compared to the other PACs. Consequently,
based on these discussions and analyses, we conclude that ozone was not the cause of the
decreasing trend of these PACs in the XAD-PASs.
**What Can Explain Decreasing Trends in the Amounts Accumulated in PASs?** After
rejecting all three of our hypotheses, we can formulate a number of constraints on any further
potential explanations. We find that the phenomenon is affecting different PACs to a different





extent, but this extent is neither related to the PACs' volatility, e.g., as expressed through the
$K_{XAD/air}$, nor to their relative reactivity with photo-oxidants. We further find that the
phenomenon appears to be very similar in calibration studies started in early winter and early
summer in a region with a strongly seasonal climate. This suggests that the strength of the
process causing this phenomenon cannot be strongly affected by season. All three of our
hypotheses relied on a seasonally variable process: PAC particle concentrations peak in winter,
temperature and therefore evaporative loss potential is highest in summer, and photo-oxidant
concentrations and degradative loss potential are also highest in summer.
We used additional model simulations (Zhang and Wania, 2012) to further shed light on this
issue. For 18 PACs we determined the combination of $k_{sorb}$ and $k_d$ values that resulted in the
best fit between model-predicted uptake curves and the uptake curves measured during the
Toronto experiment ($R^2 > 0.4$). The fits are shown in Figure S10 and the fitted parameters are
summarized in Table S10. The $k_{sorb}$ values range between 2,500 and 300,000 day$^{-1}$, with higher
values for more volatile PACs, as had been observed previously.[62] This range encompasses the
range of $k_{sorb}$ values previously estimated for polychlorinated biphenyls (10,000 to 80,000 day$^{-1}$
).[62] The $k_d$ values range from extremely low values for benzo[b]fluoranthene to 0.073 day$^{-1}$
for triphenylene, the latter corresponding to a half-life on the order of 9 days. For many of the
PACs the fitted $k_d$ values indicate a loss process with a half-life ranging from 10 days (2-
methylfluoanthene) to 100 days (phenanthrene). We further explored whether it is possible to
obtain model results that describe the observation, if it is assumed that the sorbed PACs do not
undergo loss. Uptake curves predicted using the $k_{sorb}$ values in Table S10 and a $k_d$ of zero, i.e.,
without degradative loss, deviate strongly from the measured ones (Figure 11). In other words,
the model cannot find a best fit to the observations without a loss process.
In summary, the evidence suggests the presence of a process that results in the loss of PACs
from the XAD-resin with half-lives on the order of weeks to months. The kinetics of that loss
process is different for different PACs, with the $k_d$ values in Table 10 approximating the relative
susceptibility of different PACs. We have to concede that we presently do not know the nature
of that loss process. Microbial degradation could be a possibility, although reported relative
rates of degradation for PAHs (Ghosal et al., 2016) are not consistent with our $k_d$ values, i.e.,
PAHs with shorter biodegradation half-lives do not appear to show a higher rate of loss from
the XAD-PAS. Microbial degradation is likely to be very complex and potentially related to
microbial species and communities, as well as environment conditions. More efforts are needed
in future studies to further identify the process leading to the loss of PACs from the resin.



**413** **Is It Possible to Still Derive Useful Kinetic Information from the Calibration Experiments?**

**414** $V_{eff}$ values calculated using Eq. (1) are provided in Tables S7 and S9. Remarkably, when $V_{eff}$

**415** for 15 PAHs, 12 alk-PAHs and dibenzothiophene from the Toronto calibration experiment are

**416** plotted against deployment length, i.e., if we linearize the uptake curves from Figures 1 and S2,

**417** almost all of the data show good linearity for the first six months of deployment (most linear

**418** regressions have $R^2 > 0.90$ and $p$ values $< 0.05$, Table 2). Examples of such plots for 2-

**419** methylnaphthalene, 1-methylfluorene, and retene are shown in Figure 4, with the remainder

**420** being compiled in Figure S12. This is also true for fluoranthene and pyrene measured at the

**421** clearing site in Borden, but not at the forest site (Figure S13). As only PUF plugs were used in

**422** the AASs to sample PAHs from the gas phase in Borden, five light PAHs (i.e., naphthalene,

**423** acenaphthylene, acenaphthene, fluorene, and phenanthrene) suffered from breakthrough,

**424** yielded unreliable $C_{air}$ and therefore were excluded from the data analysis.

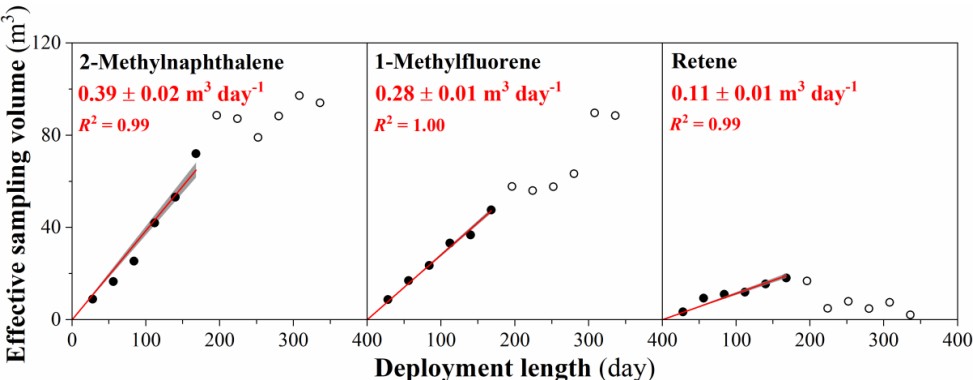

**425**

**Figure 4:** The effective sampling volume of three representative PACs throughout the 48-week

deployment period for XAD-PASs in Toronto. The markers indicate field blank-corrected

measured values, whereby data points marked with open symbols were excluded from the

regression analysis. The red lines indicate linear regressions forced through the origin. The

shaded areas represent the uncertainties of the calculated sampling rates obtained from the

uncertainties of the slopes of these regressions. Sampling rates in $m^3\ day^{-1}$ are obtained from

the slope of these regressions.

**433** As might be expected from the uptake curves, the $V_{eff}$ for most PACs is leveling-off or

**434** decreasing after six months (Figures 3, S11, and S12). Nevertheless, the linear uptake within

**435** the first half year of deployment suggests that the XAD-PAS can be used to sample PACs from

**436** the gas phase, even the most volatile one (naphthalene), as long as the deployment period does

**437** not exceed six months. Experimental sampling rates ($SR_{experimental}$) for 28 PACs, including 15





PAHs, 12 alk-PAHs, and dibenzothiophene, were estimated from the slopes of linear
regressions for the data points from the first six months of deployment and ranged from 0.05
to 0.53 $m^3$ $day^{-1}$. Uncertainties of the $SR_{experimental}$ values in the range of 2 to 15% were estimated
from the standard error of the slopes of these linear regressions. For fourteen compounds (2
PAHs, 11 alk-PAHs, and dibenzothiophene), these are the first $SR$s ever reported for the XAD-
PAS. The $SR_{experimental}$ values for fluoranthene and pyrene from the Toronto and Borden
experiments are very close, lending support to the reliability of the two experiments. The actual
sampling rates ($SR_{actual}$) for 18 PACs were obtained using the simulated no-degradation uptake
amounts in the first six months of deployment (Figure S11) and AAS data in our Toronto
experiment based on Eq. (1) (Table S11).
The $SR_{experimental}$ values in this study are lower than those reported for PAHs previously
(Armitage et al., 2013; Ellickson et al., 2017). The $SR_{experimental}$ values for various SVOCs
obtained from the same Toronto calibration experiment were also lower than those reported in
other literature (Li et al., 2023b, a). The difference between the $SR_{experimental}$ values calculated
in our study and those previously reported may be caused by the difference in ambient wind
speed (1.7, ~4.0, and ~8.0 $m·s^{-1}$ during our study, the study of Armitage et al. (2013), and the
study of Ellickson et al. (2017), respectively) and possibly also in the rates of degradative loss.
Other reasons for the relatively higher $SR$s in the study of Ellickson et al. (2017) may be due
to (1) break-through losses for more volatile PAHs during the high-volume AAS; (2) failure to
sample episodes of elevated air concentration when only sampling episodically with the AASs;
and (3) lower air concentrations at active air sampling sites than those at passive air sampling
sites.
The reliability of $SR_{experimental}$ and $SR_{actual}$ values for PACs presented here is supported by them
falling within a similar range as those for other SVOCs from the same calibration experiment
(Li et al., 2023b, a). Also, the $SR$s for PACs exhibit a consistent negative correlation with log
$K_{XAD/air}$, mirroring the pattern observed for other SVOCs (Figure 5). However, the $SR_{experimental}$
for PACs are relatively lower than those of the PCBs and other SVOCs with similar $K_{XAD/air}$
values, presumably due to the degradative loss, whereas the $SR_{actual}$ for PACs have a
relationship with $K_{XAD/air}$ that is very similar to that of the PCBs (Figure 5). A linear regression
with the log ($K_{XAD/air}$ / (L air $g^{-1}$ XAD)) at 12.2 °C yields the following relationship:
$$SR_{experimental}\ (m^3\ day^{-1}) = -0.06(\pm0.02) \log (K_{XAD/air}/L\ g^{-1}) + 0.55(\pm0.08) \qquad (2)$$

n = 30, p < 0.0005, $R^2$ = 0.35



$SR_{\text{actual}}\ (m^3\ day^{-1}) = -0.11(\pm 0.03) \log (K_{\text{XAD/air}}/\text{L g}^{-1}) + 0.90(\pm 0.17)$     (3)

$n = 18, p < 0.0005, R^2 = 0.40$

Going forward, we recommend deployment periods for the XAD-PAS to be kept shorter than
6 months, if PACs are among the targeted SVOCs. We suggest that for such shorter
deployments, the $SR_{\text{experimental}}$ reported in Table 2 can be used. For PACs not included in Table
2, their $SR_{\text{experimental}}$ and $SR_{\text{actual}}$ can be estimated using a predicted $K_{\text{XAD/Air}}$ value and Eqs. (2)
and (3). A quantitative interpretation of PAC levels in XAD-PAS deployed for periods longer
than half a year may be compromised by the variable degradation loss of PACs from the XAD-
resin. This applies retroactively also to studies that have reported PAH levels in XAD-PAS
deployed for one year; they should be considered to be semi-quantitative only (Daly et al., 2007;
Lévy et al., 2018; Choi et al., 2009; Westgate et al., 2010; Abdul Hussain et al., 2019;
Schummer et al., 2014; Schrlau et al., 2011).

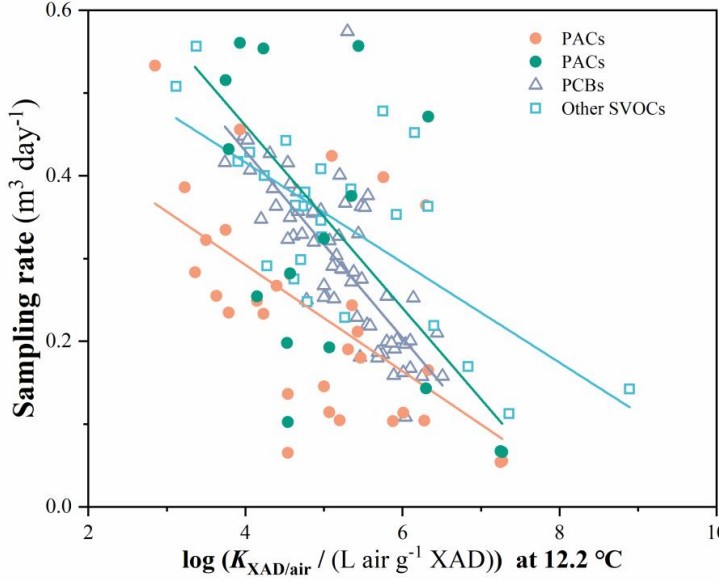


**Figure 5:** The increase of the sampling rates of 120 chemicals with an increase in volatility. The

markers in different colors indicate sampling rates of different chemical groups, and the lines

in different colors indicate linear regressions between the sampling rate and log $K_{\text{XAD/air}}$ at

12.2 °C for different chemical groups. The dark green dots and line indicate the simulated

sampling rates for PACs without degradation and linear regression for these PACs.





**Table 2** Passive Sampling Rates (m³ d⁻¹) of PACs Determined in Our Study and Reported in Literature

| Chemicals | Abbreviation | log $K_{XAD/air}$ (L air g⁻¹ XAD) at 7.9 °C | log $K_{XAD/air}$ (L air g⁻¹ XAD) at 12.2 °C | $SR_{experimental}$ in our study | $R^2$ | $SR$ in literature |
|---|---|---|---|---|---|---|
| **Two rings** | | | | | | |
| Naphthalene | NAP | 2.95 | 2.85 | 0.53 ± 0.04 | 0.97 | 1.23 (Armitage et al., 2013) 1.10 (Ellickson et al., 2017), 0.75 (Ellickson et al., 2017) |
| 2-Methylnaphthalene | 2-MeNAP | 3.33 | 3.23 | 0.39 ± 0.02 | 0.99 | |
| 1-Methylnaphthalene | 1-MeNAP | 3.46 | 3.36 | 0.28 ± 0.03 | 0.93 | |
| 2,6-dimethylnaphthalene | 2,6-DiMeNAP | 3.73 | 3.63 | 0.25 ± 0.01 | 0.99 | |
| 1,3-dimethylnaphthalene | 1,3-DiMeNAP | 3.85 | 3.75 | 0.33 ± 0.02 | 0.97 | |
| 1,5-dimethylnaphthalene | 1,5-DiMeNAP | 4.03 | 3.93 | 0.46 ± 0.02 | 0.99 | |
| 2,3,5-trimethylnaphthalene | 2,3,5-TriMeNAP | 4.34 | 4.23 | 0.23 ± 0.01 | 0.98 | |
| **Three rings** | | | | | | |
| Acenaphthylene | ACY | 3.60 | 3.50 | 0.32 ± 0.03 | 0.97 | 1.26 (Armitage et al., 2013), 0.60 (Ellickson et al., 2017) |
| Acenaphthene | ACE | 3.89 | 3.79 | 0.23 ± 0.01 | 0.99 | 0.48 (Ellickson et al., 2017) 1.05 (Armitage et al., 2013), 0.65 (Ellickson et al., 2017), 0.60 |
| Fluorene | FLU | 4.26 | 4.15 | 0.25 ± 0.01 | 1.00 | (Ellickson et al., 2017) |
| 1-Methylfluorene | 1-MeFLU | 4.68 | 4.57 | 0.28 ± 0.01 | 1.00 | |
| Dibenzothiophene | DBT | 4.64 | 4.54 | 0.06 ± 0.00 | 0.98 | |
| Phenanthrene | PHE | 4.64 | 4.54 | 0.14 ± 0.01 | 0.99 | 0.95 (Armitage et al., 2013), 0.55 |





| | | | | | | |
|---|---|---|---|---|---|---|
| Anthracene | ANT | 4.50 | 4.40 | $0.27 \pm 0.03$ | 0.94 | (Ellickson et al., 2017), 0.70 (Ellickson et al., 2017) 0.35 (Ellickson et al., 2017), 0.25 (Ellickson et al., 2017) |
| 2-Methylphenanthrene | 2-MePHE | 5.18 | 5.07 | $0.11 \pm 0.01$ | 0.99 | |
| 1-Methylphenanthrene | 1-MePHE | 5.11 | 5.00 | $0.15 \pm 0.01$ | 0.97 | |
| 9-Methylanthracene | 9-MeANT | 5.21 | 5.10 | $0.42 \pm 0.03$ | 0.98 | |
| Retene | RET | 6.12 | 6.01 | $0.11 \pm 0.01$ | 0.99 | 0.37 (Ellickson et al., 2017), 0.25 (Ellickson et al., 2017) |
| **Four rings** | | | | | | |
| Fluoranthene | FLA | 5.47 | 5.36 | $0.24 \pm 0.01, 0.18 \pm 0.01^*$ | 1.00, 0.99* | 0.80 (Armitage et al., 2013), 0.44 (Ellickson et al., 2017), 0.50 (Ellickson et al., 2017) |
| Pyrene | PYE | 5.31 | 5.20 | $0.10 \pm 0.01, 0.19 \pm 0.02^*$ | 0.98, 0.97* | 0.74 (Armitage et al., 2013) |
| 2-Methylfluoranthene | 2-MeFLT | 5.55 | 5.43 | $0.21 \pm 0.01$ | 0.98 | |
| 1-Methylpyrene | 1-MePYE | 5.87 | 5.76 | $0.40 \pm 0.05$ | 0.95 | |
| Benz[a]anthracene | BaA | 6.40 | 6.28 | $0.10 \pm 0.02$ | 0.90 | 0.50 (Ellickson et al., 2017), 0.10 (Ellickson et al., 2017) |
| Triphenylene | TP | 6.45 | 6.33 | $0.17 \pm 0.02$ | 0.94 | 0.25 (Ellickson et al., 2017) |
| Chrysene | CHY | 6.42 | 6.30 | $0.36 \pm 0.01$ | 0.99 | 0.89 (Armitage et al., 2013), 0.35 (Ellickson et al., 2017), 0.14 |



| | | | | | | |
|---|---|---|---|---|---|---|
| | | | | | | (Ellickson et al., 2017) |
| **Five rings** | | | | | | |
| Benzo(ghi)fluoranthene | BghiF | 6.00 | 5.88 | 0.10 ± 0.01 | 0.97 | |
| Benzo[b]fluoranthene | BbF | 7.39 | 7.27 | 0.06 ± 0.00 | 0.99 | 0.10 (Ellickson et al., 2017) |
| Benzo[k]fluoranthene | BkF | 7.37 | 7.25 | 0.05 ± 0.01 | 0.93 | 1.23 (Armitage et al., 2013), 0.15 (Ellickson et al., 2017) |

* Data for the clearing site from the Borden experiment. The originally calculated sampling rates are for the long version XAD-PAS, i.e., the length of XAD-2 mesh cylinders is 20 cm; thus, in this table, they were converted to the sampling rates of short version XAD-PAS (10 cm length) by dividing by two (Zhang et al., 2012).

**Acknowledgement**

We are grateful to Rudy Boonstra for logistical help during field work in Toronto, and Bondi Gevao for help sampling in Borden. Financial support from a Grant and Contribution Agreement (GCXE20S008) with Environment and Climate Change Canada and a Connaught scholarship to YL is gratefully acknowledged. The work at Borden was supported by a grant from the Canadian Foundation for Climate and Atmospheric Sciences (CFCAS).

**Code/Data availability**

All data generated for this project are contained in the supplement.

**Author contribution**

YL and FZ performed measurements and analyzed samples from the Toronto experiment, YS performed measurements and analyzed samples from the Borden experiment, both under the supervision of YDL. CS analysed the atmospheric particle samples. YL performed the ozone flow tube experiment with guidance by ZZ and JPDA. YL and FZ performed the model simulations and YL interpreted the data and wrote the manuscript under guidance by FW. HH coordinated the project. All authors reviewed the manuscript.

**Competing interests**

The authors declare no competing interests.

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
