# Peer review of "Uptake Behavior of Polycyclic Aromatic Compounds during Field"

_EGUsphere, 2023_

## Author Comment (AC1)

**Response to Reviewer 1**

*This manuscript makes a strong contribution to advancing our understanding of passive air sampling by critically evaluating the existing paradigms of sampler deployment and, importantly, identifying a strong source of uncertainty that is being introduced into the data. The overview of the state of passive air sampling is accurate and correctly captures the uncertainties in passive sampling sorbents, configurations, and uptakes that impact PAC quantification by PAS.*

*The authors take a step-by-step approach in testing a series of hypotheses - this structure is very clear, and very comprehensively integrates the results from the different methods that were combined in the study. The scientific methods and interpretation are very good, and only a few points require further clarification.*

> We appreciate the reviewer's endorsement of our work.

*Specific points for the authors to clarify:*

*Please clarify hypothesis 1 "Can Uptake of Particle-bound Compounds Explain Decreasing Trends in the Amounts Accumulated in PASs?" While I fully agree with the methodology used to test the hypothesis and the conclusions the authors have made with respect to this hypothesis, I question how this phenomenon could lead to decreases in chemicals in the samplers over time. I rather think this would lead to irregular/non-linear uptake. Does this hypothesis assume that particles can be "blown-off" the sorbents in over time?*

> The reviewer is correct that the uptake of particle-bound compounds does not provide an explanation for the amount of a chemical quantified in the sampler sorbent decreasing with deployment time. This hypothesis is rather suggesting that because the uptake of particles can be influenced by the size, type, and concentrations of particles in air and the wind exposure of a PAS during deployment, it could result in inconsistent and unpredictable uptake of PACs, including in uptake curves that could be suggestive of compound loss from the sampler sorbent. We agree that this was not clearly articulated. We have rephrased the introduction of that hypothesis as follows:

The first hypothesis posits that the observed uptake behaviour was caused by the  uptake and accumulation of atmospheric particles in the XAD-PAS. This hypothesis is based on previous studies showing that some PASs do not efficiently block wind from carrying particle-bound substances to the sorbent, as is, e.g., observed in the case of the PUF-PAS with double bowl shelters (Wania and Shunthirasingham, 2020; Chaemfa et al., 2009; Bohlin et al., 2014b; Harner et al., 2013). Even though there is currently no evidence of the uptake of particle-bound SVOCs in the XAD-PAS, if such uptake were to occur, the accumulated amount of a chemical in the XAD-PAS would be the sum of the amount taken up from the gas phase and the amount derived from sampled particle phase. **While this admittedly cannot explain why the amount of a chemical quantified in the sorbent would decrease with deployment time, it could result in inconsistent and unpredictable uptake of PACs, because such**  uptake could be influenced by the size, type, and concentrations of particles in air and the wind exposure of a PAS during deployment. **Some such inconsistent uptake could conceivably by suggestive of compound loss from the sampler sorbent.**

*Please consider adding more specificity to the naming of the hypothesis related to degradation (hypothesis 3) to make it clear that this is related to photooxidants, largely ozone. In fact, the outcome of the study seems to be that some other unknown degradation pathway is contributing to the observed patterns, but that aspect is not tested under the 3rd hypothesis.*

> We agree with the reviewer and have changed the name of hypothesis 3 to "Can Reactions with Photooxidants Explain Decreasing Trends in the Amounts Accumulated in PASs?"

---

## Author Comment (AC2)

**Response to Reviewer 2**

*General comments*
*The manuscript discusses the uptake of polycyclic aromatic compounds by a specific passive air sampler for deployments lasting up to 1 year. Contrary to expectations, several compounds did not remain in the linear uptake phase; several PAHs either plateaued, or decreased after about 6 months deployment. The behaviour observed for PAHs was in contrast to results for other semi-volatile organic compounds. Three possible reasons were explored (i) impact of particle-bound PACs; (ii) degradation of PAC by ozone and (iii) equilibration of certain PACs.*
*None of these were concluded to be responsible for the observed observations.*
*Overall, the manuscript is well written, and includes good data, figures and tables.*
> We appreciate the positive feedback.

*There is a lack of additional creativity beyond the 3 reasons investigated by the authors. Let's try a few:*

1. *What amount of XAD is oxidized, and thus losing sorption capacity during the year?*

2. *The curves depicted in Figure 1 might indicate competition for sorptive sites. And the PACs might simply be replaced by stronger sorbing compounds.*

> Re 1: There is little reason to expect that oxidation of the XAD should result in a loss of its sorption capacity for PACs, unless that oxidation would result in a massive loss of surface area. On the contrary, one might expect oxidation of XAD to lead to a functionalization of its surface, i.e., the introduction of oxygen-containing functional moieties such as carbonyl, hydroxy and carboxylic acid groups. A more polar surface could be expected to sorb PACs more strongly than the original polystyrene-divinylbenzene resin, because of the addition of possible polar interactions between sorbent and sorbate.

> Re 1 and 2: A loss of sorption capacity and competitive sorption are simply alternative reasons for a "limited sorbent uptake capacity" and therefore can be subsumed under the second hypothesis we explore. Whatever the underlying reasons for an insufficiently large sorptive capacity, the extent of loss of a PAC from the sampler would be expected to be related to compound volatility, which is not observed. We have added the following sentence to the introduction of the second hypothesis:

"A variant of this hypothesis is that something is causing a decrease in the sorption capacity of the XAD-resin over time, i.e. the aging of the resin or competition by other sorbates."

*Specific comments*
*L66 – very detailed list what is the difference between urban and regions with high traffic density?*
> Not all regions with high traffic density are in urban areas. Some highways with high traffic volume are in rural areas, if they connect major population centres.

*L 149 – give better reason why ozone was chosen (NOx or OH)*
> We are not sure why the reason we provide in the paragraph introducing the third hypothesis are not good enough. With regard to $NO_X$ we write: "While nitrogen dioxide may also react with some PAHs, either no reactions were found (Grosjean et al., 1983; Pitts et al., 1980), or only negligible percentages of PAHs were observed to react with nitrogen dioxide (Tokiwa et al., 1981)." With regard to OH we write: "Even though gaseous and sorbed PAHs can react with OH radicals (Brubaker and Hites, 1998; Atkinson and Arey, 2007; Esteve et al., 2004, 2006; Bedjanian et al., 2010), such reactions are less likely to occur within the dark environment of a XAD-PAS housing."

Formation of OH requires sunlight and the atmospheric lifetime of OH radicals is very short, on the order of a second or less. It is a dark within the XAD-PAS housing, and it is not very likely that many OH radicals would survive long enough to diffuse to the sorbent to react with the sorbed PACs. We have added the phrase: ", because of the short atmospheric lifetime of OH radicals" to make this clearer.

*L193 – reference for chosen air boundary later?*

It is a commonly used air boundary layer thickness in the model by Zhang and Wania (2012). We now refer to two papers have used this thickness during simulations "Zhang and Wania, 2012  Li et al., 2023b"

*Figure 3 – This is supposed to simulate an equilibration experiment, right? Make that explicit in the caption*

We modified the caption to Figure 3 as follows " Predicted uptake curves for a **volatile** chemical **reaching equilibrium between XAD and gas phase, i.e.,** with a log ($K_{\text{XAD/air}}$ / (L air g$^{-1}$ XAD) value of 2.69 at 20 °C, during calibration experiments starting in June and November, while assuming air concentration and temperature variability typical for Southern Ontario."

*Figure 5 would benefit from some descriptive statistics – presumably all compounds but PACs, and PACs on their own*

Figure 5 displays four relationships between the empirically determined sampling rates (SRs) and the estimated sorption constant to XAD-resin. Descriptive statistics for those relationships for the PCBs and other SVOCs have been provided in earlier publications (Li et al., 2023a, b). Descriptive statistics for the relationships for the PACs are provided in the text (equations 2 and 3).

We suggest that the relationship for PCBs be used to estimate sampling rates for PCBs (Li et al., 2023b) and that the relationship from Li et al., 2023a be used for estimating SRs for SVOCs other than PCBs and PACs. As such, we do not think it is advisable to provide another relationship that combines the SRs for all compounds other than the PACs.

*Technical corrections*
*L145 – sandwich*

We prefer to keep using the plural "sandwiches" as we used more than one PUF-XAD-PUF sandwich during our active air sampling.

*L335 – Thus it should?*

The original formulation is also grammatically correct.